# Factors impacting adherence to an exercise-based physical therapy program for individuals with low back pain

Bahar Shahidi[1]*, Jennifer Padwal[1], Euyhyun Lee[2], Ronghui Xu[2,3,4], Sarah Northway[1,5], Lissa Taitano[5], Tiffany Wu[5], Kamshad Raiszadeh[5]

**1** UCSD Department of Orthopaedic Surgery, La Jolla, CA, United States of America, **2** UCSD Altman Clinical and Translational Research Institute- Biostatistics, La Jolla, CA, United States of America, **3** UCSD Department of Family Medicine and Public Health, La Jolla, CA, United States of America, **4** UCSD Department of Mathematics, La Jolla, CA, United States of America, **5** SpineZone Medical Fitness, San Diego, CA, United States of America

* bshahidi@health.ucsd.edu

**Data Availability Statement:** Data underlying the results presented in the study are available in the Supporting Information files associated with the submission.

## Abstract

### Background/Objective

Exercise-based rehabilitation is a conservative management approach for individuals with low back pain. However, adherence rates for conservative management are often low and the reasons for this are not well described. The objective of this study was to evaluate predictors of adherence and patient-reported reasons for non-adherence after ceasing a supervised exercise-based rehabilitation program in individuals with low back pain.

### Design

Retrospective observational study.

### Methods

Data was retrospectively analyzed from 5 rehabilitation clinics utilizing a standardized exercise-based rehabilitation program. Baseline demographics, diagnosis and symptom specific features, visit number, and discontinuation profiles were quantified for 2,243 patients who underwent the program.

### Results

Forty-three percent (43%) of participants were adherent to the program, with the majority (31.7%) discontinuing treatment prior to completion due to logistic and accessibility issues. Another 13.2% discontinued prior to the prescribed duration due to clinically significant improvements in pain and/or disability without formal discharge evaluation, whereas 8.3% did not continue due to lack of improvement. Finally, 6.0% were discharged for related and unrelated medical reasons including surgery. Individuals diagnosed with disc pathology were most likely to be adherent to the program.

**Funding:** This project was funded by the American Physical Therapy Association Foundation for Physical Therapy Research Magistro Family Foundation Grant awarded to BS. The Clinical Translational Research Institute (CTRI) is partially supported by the National Institutes of Health, Grant UL1TR001442 of CTSA funding. The funders had no role in study design, data collection, analysis, decision to publish, or preparation of the manuscript. The authors have no conflicts to declare.

**Competing interests:** The authors have declared that no competing interests exist.

**Abbreviations:** LBP, Low back pain; BMI, Body mass index; MVC, Maximum Voluntary Contraction; NSAID, Nonsteroidal anti-inflammatory drug; ODI, Oswestry Disability Index; EQ5D, EuroQol 5D.

## Limitations

This study was a retrospective chart review with missing data for some variables. Future studies with a prospective design would increase quality of evidence.

## Conclusions

The majority of individuals prescribed an in-clinic exercise-based rehabilitation program are non-adherent. Patient diagnosis was the most important predictor of adherence. For those who were not adherent, important barriers include personal issues, insufficient insurance authorization and lack of geographic accessibility.

## Introduction

Low back pain (LBP) is a debilitating and costly condition affecting 65–85% of the United States population during their lifetime [1–3]. Although acute LBP is thought to be self-limiting in the short term, with most symptoms resolving within 3 months of onset, recurrences and progression to chronic LBP are observed in 24–80% of cases [4]. Improving strength and stability of the trunk musculature through therapeutic exercise is a common physical rehabilitation goal in this population, and is thought to improve functional outcomes by both facilitating hypertrophy of the supporting paraspinal musculature and decreasing or preventing commonly observed maladaptive physiological changes such as muscle atrophy and fatty infiltration [5–9]. However, despite the observation of short-term improvements in response to standard rehabilitation programs, these improvements often do not persist in the long term. One possible reason for this is that the most commonly prescribed exercise doses and durations may be insufficient to induce physiological change in the affected tissues [10–13]. Indeed, most studies demonstrating exercise-induced changes in the form of muscle growth required treatment durations longer than are typically prescribed: an average of 16 weeks [10, 14]. Further, when implemented at these durations, they resulted in not only short-term, but also long term improvements in functional outcomes [12, 15] in addition to reductions in healthcare resource utilization by 87% after 1 year [11, 16, 17]. Despite evidence supporting longer treatment durations, the number of visits utilized for exercise-based rehabilitation remain lower, averaging 8–12 visits over a 6–8 week period [18].

One reason for the discrepancy in treatment volume is that adherence, or attendance to supervised treatment, for sustained rehabilitation programs in clinical practice settings is often low, with rates varying as widely as 15–87% [19, 20]. A systematic review [21] of literature published between 1998 and 2014 on adherence to therapeutic exercise interventions yielded only 3 studies including patients with LBP [22–24]. Additionally, literature investigating adherence for supervised exercise programs lasting longer than 6 weeks is absent in this population, and of those reporting shorter-term adherence rates (<6 weeks), the largest sample size reported was 170 participants, with broad ranges of non-adherence (51–87%) [25, 26]. Adherence for these studies was based on self-reported time spent performing a home exercise program, making comparisons to supervised rehabilitation difficult, although adherence rates are also similarly wide (15–70%) [19, 20]. Wide ranges of adherence have also been observed in other musculoskeletal conditions such as hip or knee osteoarthritis, with levels ranging between 26–52% [27, 28].

These low adherence rates often go unrecognized in the literature given that many clinical trials likely over-represent adherence due to selection bias and resources being allocated to patient follow-up and retention as compared to normal clinical practice. Additionally, logistic limitations such as lack of insurance coverage and accessibility restrictions have been shown to affect trial enrollment and may reduce selection of populations with restricted resources [29, 30]. Indeed, the lack of geographic accessibility and concern over financial burden have increasingly been shown to disproportionately impact individuals of low socioeconomic status, resulting in widening gaps in healthcare disparities [31]. High medical comorbidity and medical safety is also a consideration in individuals with back pain, as this has been shown to influence clinical outcomes and ability to return to function possibly due to inability to safely tolerate an exercise-based program [32]. Conversely, publication bias against pragmatic trials with high levels of loss to follow up and poor resolution for evaluating clinical efficacy contributes to underreporting of the prevalence of low adherence in these populations and settings [33]. Indeed, a recent Cochrane Review reported that of 381 studies of exercise-based rehabilitation programs for chronic pain conditions, adherence could not be assessed in any review, and healthcare use/attendance was not reported in any of the reviews [34]. As such, it is important to provide data on adherence in typical clinical practice settings, and to evaluate the factors that contribute to the low rates observed given that it is potentially a key barrier to achieving optimal therapeutic efficacy [35, 36].

The purpose of this investigation is to evaluate adherence for a standardized 20 visit (10 weeks) supervised in-clinic exercise-based rehabilitation program taking place in an outpatient physical therapy setting in a large group of patients with LBP. Furthermore, factors predicting high adherence as well as reasons for becoming non-adherent are explored. These data will provide key insight into improving care accessibility and will help identify targets for optimizing patient retention and treatment outcomes.

## Methods

### Patient characteristics

This project was approved by the local ethical review board (Western IRB #1180578) and a waiver of consent was obtained due to the nature of the de-identified data. Patient data was collected retrospectively from 5 outpatient physical therapy clinics in the greater San Diego area for patients who received treatment at one or more of these clinics between November 2015 and June 2017. Patient characteristics data that have been demonstrated to be impactful in clinical outcomes were collected. Specifically, age [37], sex [37, 38], body mass index (BMI), baseline medication usage [39], smoking history [38], diabetes [38], low back pain-specific diagnosis [37], pain visual analogue scale (VAS) [4], low-back pain related disability [40] from the Oswestry Disability Index (ODI), and quality of life from the EuroQol-5D (EQ5D) questionnaire have been shown to influence outcomes and were extracted. Baseline isometric lumbar extension strength (in ft*lbs) was collected during a maximum voluntary contraction (MVC) measured on a MedX isokinetic dynamometer (Baseline Exercise). Low back pain diagnoses were categorized based on ICD-9 codes associated with the following conditions: nonspecific LBP, degenerative disc disease/disc herniation, lumbar stenosis, and spondylolisthesis. These diagnostic categories are consistent with previous literature in large systematic reviews and Cochrane databases [10, 41, 42]. Based on previously described diagnostic criteria, patients were categorized in the lumbar stenosis category if they had 1) neurogenic claudication and/or radicular leg symptoms, or 2) confirmatory cross-sectional imaging showing lumbar spinal stenosis at one or more levels [43]. Patients were categorized as having degenerative spondylolisthesis if they had one or more of the following: 1) neurogenic claudication or

radicular leg pain with associated neurological signs, 2) spinal stenosis seen on cross-sectional imaging, or 3) degenerative spondylolisthesis of any grade seen on standing lateral radiographs. Patients were categorized as having disc disease/herniation if they had 1) lumbar radiculopathy and a disc herniation or pathology at a corresponding level and laterality as verified on imaging if available [44, 45]. Patients that did not have a specific diagnosis associated with supportive imaging or clear symptomology were categorized as having nonspecific LBP.

## Exercise protocol

The rehabilitation protocol consisted of a standardized high intensity rehabilitation exercise program prescribed and supervised by licensed physical therapists as previously described in detail [46]. Briefly, the program consisted of a recommendation of 20 visits at 2 visits/week (a duration of 10 weeks), lasting approximately 45 minutes including lumbar extension resistance exercises performed on a MedX isokinetic dynamometer machine. Exercise dose was prescribed based on a Maximum Voluntary Contraction (MVC) and targeted 60–80% of that maximal effort for 15–20 repetitions. Exercise was advanced in subsequent visits by 5–10% of the exercise load once the patient was able to perform >20 repetitions. If they were able to reach >10 repetitions but <20 repetitions, their exercise load remained the same at their next visit. If they were unable to reach 10 repetitions, their exercise load was decreased 5–10% at their next visit.

## Measurement of adherence

The total number of visits completed was used as a measure of adherence to the prescribed program. Because the instructions provided upon prescription of the program recommended that patients complete at least 75% of the prescribed 20 visits, patients were classified as "completers" if they completed at least 16 of the 20 prescribed visits or were formally discharged due to symptom resolution. Patients who completed 15 or fewer visits and were not successfully discharged from the program were considered "non-completers". These guidelines were primarily based on the concept that muscle hypertrophy changes require at least 6–7 weeks of consistent resistance training to elicit physiological adaptation [47]. Patients who did not complete the program as prescribed were provided a discharge questionnaire indicating their reason for discontinuation of care. Reasons for discharge were categorized based on the most commonly observed reasons provided by the patient, and included a) logistic issues (personal issues, lack of sufficient insurance coverage/authorization, lack of geographic accessibility), b) medical discharge (related or unrelated health issues or progression to surgery), c) clinically important improvement in pain (>2/10 improvement on the numeric pain rating scale) [48] and/or disability (>10/100 improvement on the Oswestry Disability Index) [49] without official discharge, or d) lack of improvement/did not like the program.

## Statistical analysis

Continuous and categorical variables were summarized as mean (SD) and count (percentage), respectively. In order to evaluate patient factors that predicted adherence, univariate logistic regressions were performed with program completion as a binary dependent variable to evaluate the significance of each predictor of interest. Independent predictors included age, sex, BMI, smoking history (yes or no), diabetes (yes or no), presence of radicular symptoms (yes or no), frequency of narcotic usage (None, <1/day, 1-2/day, 3+/day), frequency of Non-Steroidal Anti-inflammatory Drug (NSAID) usage (none, <1/day, 1-2/day, 3+/day), and LBP diagnosis (nonspecific, disc herniation, spondylolisthesis, stenosis). For the narcotic and NSAID use variables, non-use (None) was considered the reference category, and for the variable of LBP

diagnosis, non-specific LBP was considered the reference category. Predictors with a univariate p-value of <0.2 were entered into a multivariable model. Subsequently, a multivariable logistic regression model was built using these variables to evaluate whether there were specific patient characteristics that predicted adherence while adjusting for other variables. Missing data were handled using pairwise deletion (patients were only removed from analysis only for multivariate, but not univariate analyses), with no replacement (no imputation was utilized to fill missing values).

Reasons for non-adherence based on patient charts, provider correspondence, and discharge questionnaires, were categorized and reported as count (percentage) as a function of the total number of participants included in the study (adherent and non-adherent). All statistical analyses were performed in R (V.3.6.1, R Core Team (2019). R: A language and environment for statistical computing. R Foundation for Statistical Computing, Vienna, Austria).

## Results

### Patient characteristics

Of the initial sample of 2,749 subjects, 573 patients were excluded; 2 patients were excluded due to having documented ages of <0 or >100, 67 patients were excluded due to having a documented BMI that was non-physiological and likely entered in error, 218 patients had a primary diagnosis that did not fall into the preidentified diagnostic categories (e.g. fracture following trauma, scoliosis), and 288 patients were missing one or more data points related to individual visits. After these exclusions, data for the remaining 2,243 patients (81.5%) were analyzed. Overall, 958 (42.7%) participants completed at least 16 of the 20 prescribed treatment sessions and were considered adherent. The average (SD) number of visits for individuals who were adherent was 17 (5) visits, and the average number of visits for individuals who were considered non-adherent was 6 (4) visits (Table 1).

### Predictors of adherence

Of the 13 predictor variables, age, primary diagnosis, baseline VAS, baseline ODI, and baseline EQ5D scores demonstrated p-values of <0.2 in univariate analyses (Table 2). When these resulting variables were used to build the multivariate logistic regression model, primary diagnosis remained as a significant predictor of adherence (p = 0.02), with individuals with a primary diagnosis of disc pathology being the most likely to be adherent to the program (OR (95% CI) 1.47 (1.16,1.86), p = 0.002) as compared to patients with nonspecific low back pain (reference group), spondylolisthesis, or stenosis (Table 3).

### Factors contributing to non-adherence

From the total cohort, 1,285 participants were considered non-adherent. Of those, 710 (31.7%) were unable to continue treatment due to logistic reasons. The most common logistical reason for discontinuing treatment was personal issues (457 (20.3%)), followed by lack of accessibility due to no insurance authorization (168 (7.4%)) and lack of geographic accessibility (85 (3.7%)). Two hundred ninety-six (13.2%) participants experienced clinically significant improvements in pain and/or disability prior to the 16th visit but did not undergo formal discharge evaluation, and 188 (8.4%) participants reported that they discontinued specifically because they felt no improvement or did not like the program. One hundred thirty-five participants (6.0%) were discharged prior to completion of treatment due to medical reasons. Of the patients who were medically discharged, 99(4.4%) were discharged due to unrelated medical problems (e.g. other unrelated surgery, cardiac issues), 21(0.9%) were discharged by their

**Table 1. Patient characteristics.** Data are means (SD) unless otherwise indicated.

| Mean (SD) or % | Non-adherent | Adherent | Overall |
|---|---|---|---|
| **Age (years)** | 54.02 (17.23) | 55.06 (16.55) | 54.50 (16.92) |
| **Sex, M/F (%)** | 44.8/55.2 | 42.7/57.3 | 43.1/56.9 |
| BMI (kg/m$^2$) | 27.64 (5.28) | 27.76 (5.44) | 27.70 (5.36) |
| **Smoking History, no/yes (%)** | 93.1/6.9 | 93.5/6.5 | 93.1/6.9 |
| **Diabetes Diagnosis, no/yes (%)** | 90.9/9.1 | 90.0/10.0 | 90.3/9.7 |
| **Radiculopathy Diagnosis, no/yes (%)** | 43.0/57.0 | 40.7/59.3 | 41.9/58.1 |
| **Baseline Narcotic Usage (%)** | | | |
| None | 60.5 | 61.0 | 60.6 |
| <1/day | 14.5 | 15.6 | 14.8 |
| 1-2/day | 14.8 | 14.1 | 14.8 |
| 3+/day | 10.2 | 9.3 | 9.9 |
| **Baseline NSAID Usage (%)** | | | |
| None | 44.3 | 41.0 | 43.1 |
| <1/day | 19.7 | 20.9 | 20.1 |
| 1-2/day | 23.3 | 25.4 | 24.1 |
| 3+/day | 12.8 | 12.7 | 12.7 |
| **Primary Diagnoses (%)** | | | |
| Nonspecific LBP | 62.3 | 56.0 | 58.7 |
| Disc herniation | 14.3 | 19.7 | 16.8 |
| Spondylosis/spondylolisthesis | 6.3 | 6.4 | 6.9 |
| Stenosis | 17.1 | 17.9 | 17.5 |
| **Baseline VAS (mm)** | 54.39 (21.83) | 52.47 (21.66) | 53.56 (21.77) |
| **Baseline ODI (%)** | 29.52 (15.79) | 28.30 (15.34) | 28.96 (15.59) |
| **Baseline EQ5D (points)** | 0.71 (0.14) | 0.72 (0.13) | 0.72 (0.14) |
| **Baseline Exercise (ft*lb)** | 816.30 (476.91) | 838.11 (470.50) | 827.53 (473.64) |
| **Number of Visits** | 6.28 (4.40) | 17.55 (4.54) | 11.15 (7.15) |

SD: Standard Deviation; M: Male; F: Female; NSAID: Nonsteroidal Anti-inflammatory Drug; LBP: Low Back Pain; VAS: Visual Analogue Scale; ODI: Oswestry Disability Index; EQ5D: EuroQol-5 Dimension; ft*lb: foot-pound.

referring physician for additional work-up or alternative therapy, and 15(0.7%) were discharged to continue with spinal surgery for their condition.

## Discussion

This is the largest study to our knowledge to report adherence levels in a long term (>8 weeks) supervised exercise-based rehabilitation program in a cohort of individuals with LBP, and to demonstrate diagnosis-specific differences in adherence within the patient population. Additionally, this is the only study to investigate reasons for non-adherence to supervised exercise-based rehabilitation in a quantitative manner, although some data is reported on adherence to musculoskeletal physical therapy in women with LBP without specifying an exercise component [50]. We found that just under half of participants prescribed a sustained exercise-based rehabilitation program were adherent to the recommended prescription, with most non-adherent patients discontinuing due to logistic reasons. A smaller proportion of patients either discontinued early due to improved symptoms without returning for formal discharge evaluation, or conversely did not improve enough to complete the full treatment as prescribed. Less than 5% of the cohort was unable to complete the program due to medical issues, and less than 1% of the cohort crossed over to surgery during the prescribed treatment period, suggesting

**Table 2. Univariate logistic regression with the program completion as the dependent variable.** Variables with asterisks were included in the multivariable model.

| | Odds Ratio | 95% CI | $X^2$ value | p-value |
|---|---|---|---|---|
| Age (years)* | 1.00 | (0.99, 1.01) | | 0.13 |
| Female sex | 1.09 | (0.93, 1.28) | | 0.30 |
| BMI (kg/m$^2$) | 1.00 | (0.99, 1.02) | | 0.58 |
| Smoking History (yes/no) | 0.93 | (0.67, 1.28) | | 0.65 |
| Diabetes (yes/no) | 1.10 | (0.84, 1.45) | | 0.48 |
| Radiculopathy (yes/no) | 1.10 | (0.94, 1.29) | | 0.24 |
| Baseline Narcotic Usage: Reference = none | | | $X^2$ = 1.19 | 0.76 |
| <1/day | 1.07 | (0.85, 1.35) | | 0.57 |
| 1-2/day | 0.95 | (0.75, 1.2) | | 0.64 |
| 3+/day | 0.91 | (0.69, 1.20) | | 0.50 |
| Baseline NSAID Usage: Reference = none | | | $X^2$ = 3.03 | 0.39 |
| <1/day | 1.15 | (0.92, 1.42) | | 0.22 |
| 1-2/day | 1.18 | (0.96, 1.44) | | 0.12 |
| 3+/day | 1.07 | (0.823 1.38) | | 0.60 |
| Primary Diagnosis:* Reference = Nonspecific LBP | | | $X^2$ = 14.94 | 0.002 |
| Disc herniation* | 1.54 | (1.23, 1.92) | | < 0.001 |
| Spondylosis/spondylolisthesis | 1.12 | (0.80, 1.56) | | 0.51 |
| Stenosis | 1.17 | (0.94, 1.46) | | 0.15 |
| Baseline VAS (points)* | 0.99 | (0.992, 0.999) | | 0.037 |
| Baseline ODI (points)* | 0.99 | (0.99, 1.00) | | 0.06 |
| Baseline EQ5D (points)* | 1.66 | (0.93, 2.97) | | 0.09 |
| Baseline Exercise (ft*lbs) | 1 | (1.00, 1.00) | | 0.29 |

BMI: Body Mass Index; NSAID: Nonsteroidal Anti-inflammatory Drug; LBP: Low Back Pain; VAS: Visual Analogue Scale; ODI: Oswestry Disability Index; EQ5D: EuroQol-5 Dimension

that unrelated or related medical issues did not substantially impact observed adherence rates in the current study. These data also highlight that an important barrier to adherence to an exercise-based rehabilitation program is lack of accessibility to care, with over 20% of non-adherent participants reporting discontinuation due to inability to obtain insurance authorization or transportation means.

**Table 3. Multivariable logistic regression analysis results with the program completion as the dependent variable.** (N = 2181; 954 adherent). Significant p-values are bolded.

| | Odds Ratio | 95% CI | $X^2$ value | p-value |
|---|---|---|---|---|
| (Intercept) | - | - | - | 0.34 |
| Age (years) | 1.00 | (0.99, 1.01) | | 0.46 |
| Primary Diagnosis: Reference = nonspecific LBP | | | $X^2$ = 9.72 | **0.02** |
| Disc Herniation | 1.47 | (1.16, 1.86) | | **0.002** |
| Spondylolisthesis | 1.08 | (0.75, 1.54) | | 0.68 |
| Stenosis | 1.16 | (0.92, 1.47) | | 0.21 |
| Baseline VAS (mm) | 0.99 | (0.99, 1.00) | | 0.19 |
| Baseline ODI (%) | 0.99 | (0.99, 1.01) | | 0.64 |
| Baseline EQ5D (points) | 1.27 | (0.54, 2.99) | | 0.59 |

CI: Confidence Interval; LBP: Low Back Pain; VAS: Visual Analogue Scale; ODI: Oswestry Disability Index; EQ5D: EuroQol-5 Dimension

In terms of diagnosis-specific predictors of adherence, we demonstrated that patients who have been diagnosed with disc herniation were 47% more likely to complete the program compared to those with non-specific LBP. These patients were also 39%, and 31% more likely to complete the program than patients diagnosed with spondylolisthesis or stenosis respectively. Although there is no prior data reporting diagnosis-specific differences in adherence levels, the natural history of improvement in disc herniation has been reported to be shorter than other more degenerative conditions [51], which may partly explain the high adherence rates in this population. Interestingly, other demographic characteristics such as age, sex, and comorbidities (i.e. smoking and diabetes) were not related to adherence, indicating that these characteristics do not limit individuals from participating in treatment in our cohort. This information can potentially inform rehabilitation clinics in identifying individuals who are at risk for being non-adherent and may encourage clinicians to consider alternative retention strategies or modified programs for individuals with specific diagnoses. Understanding barriers to adherence may also provide direction in targeted strategies for reducing patient drop out due to financial and geographical barriers. For example, providing mobile alternatives (e.g. online- or telehealth-based programs) that reduce healthcare system and patient financial burden, as well as provide flexibility in patient engagement is a feasible alternative [52]. It also indicates that insurance-based constraints on visit number may need re-evaluation relative to value of care. Finally, it will provide useful information to design future clinical trials investigating efficacy.

## Barriers to adherence

There has been a recent focus on identifying barriers to adherence and accessibility to care for management of musculoskeletal and other chronic conditions, however, there continues to be a sparsity of data elucidating these issues. Qualitative studies assessing views of patients with LBP on barriers to physical activity and exercise have reported that pain, lack of time, and difficulty with integration into daily life were primary factors limiting adherence [53], with supervision and support by professionals positively influencing adherence [53, 54]. Similarly, studies using various interventions to improve adherence have supported the concept of supervised sessions and motivational strategies [21]. These data are, in part, consistent with our finding that personal factors were a large contributor to non-adherence for those who discontinued care. Another barrier to adherence that we identified was lack of financial and geographic accessibility to care, including lack of sufficient health insurance coverage for the prescribed treatment duration. Although health insurance related factors and geographic location have not been previously identified as barriers to adherence, one recent study investigating risk factors for physical therapy visit cancellations or "no-shows" reported that insurance type and clinic location were significant predictors of not showing up for physical therapy visits [55], and these two factors are repeatedly cited as barriers to participating in clinical trials research [30]. Although "no-shows" and adherence are not identical constructs, and as stated previously, clinical trials and clinical practice behave differently, these findings highlight the potential impact of socioeconomic factors on care accessibility and compliance.

## Limitations

There are several limitations to this study. First, this is a retrospective study design and therefore causality cannot be inferred from these data. Additionally, there are a number of factors that have been previously reported in the literature to impact clinical outcomes (e.g. psychosocial variables) that may also influence adherence, that were not collected in this dataset. The number of patients with complete data varied for each variable in the regression-based

prediction model, and therefore patients that were missing data for the variables included in the final model were not accounted for (506 patients). However, the difference in N-size between the univariate analyses and multivariate analyses was less than 20% of the full cohort. Similarly, because a large proportion of participants were not adherent, clinical outcomes such as pain levels and functional status in response to treatment were not consistently available to concurrently evaluate treatment efficacy, and the ability to connect patient characteristics with reasons for non-adherence was limited because of the anonymous nature of the response data. Despite these limitations, our levels of missing data were either equivalent to, or lower than many studies investigating adherence. For example, studies measuring session attendance reported over 50% of their outcome data missing, although these studies reported missing data for longer term follow up beyond termination of the prescribed exercise program [28]. Finally, although the investigated program was a supervised rehabilitation program, program compliance—or the actual completion of prescribed activities outside of the supervised component of the program (e.g. home exercises), is not considered or measured in this investigation. Further research is needed to distinguish the impact of adherence and compliance on clinical outcomes.

## Conclusion

The majority of participants with LBP undergoing a long-term supervised exercise-based rehabilitation program designed were non-adherent using strict criterion for program completion. However, a substantial proportion of those that were considered non-adherent reported clinically important symptom and/or disability reduction. Individuals diagnosed with disc pathology were more adherent to the program than those with other diagnoses. Most individuals who initiated but did not complete the program did so due to logistical problems such as personal issues, lack of sufficient insurance authorization and geographic accessibility to a clinic. Future research is required to target effective methods for influencing these factors to improve adherence and optimize long-term treatment efficacy in this treatment setting.

## Supporting information

**S1 Data.**
(CSV)

## Author Contributions

**Conceptualization:** Bahar Shahidi, Jennifer Padwal, Kamshad Raiszadeh.

**Data curation:** Bahar Shahidi, Jennifer Padwal, Sarah Northway, Lissa Taitano, Tiffany Wu, Kamshad Raiszadeh.

**Formal analysis:** Jennifer Padwal, Euyhyun Lee, Ronghui Xu, Sarah Northway, Lissa Taitano, Tiffany Wu.

**Funding acquisition:** Bahar Shahidi, Jennifer Padwal, Kamshad Raiszadeh.

**Investigation:** Bahar Shahidi, Kamshad Raiszadeh.

**Methodology:** Bahar Shahidi, Jennifer Padwal, Ronghui Xu, Lissa Taitano, Tiffany Wu, Kamshad Raiszadeh.

**Project administration:** Bahar Shahidi, Ronghui Xu, Kamshad Raiszadeh.

**Resources:** Bahar Shahidi, Kamshad Raiszadeh.

**Software:** Euyhyun Lee.

**Supervision:** Bahar Shahidi, Kamshad Raiszadeh.

**Validation:** Bahar Shahidi, Ronghui Xu, Sarah Northway, Lissa Taitano, Tiffany Wu, Kamshad Raiszadeh.

**Visualization:** Euyhyun Lee.

**Writing – original draft:** Bahar Shahidi, Jennifer Padwal, Euyhyun Lee, Ronghui Xu, Sarah Northway, Lissa Taitano, Tiffany Wu, Kamshad Raiszadeh.

**Writing – review & editing:** Bahar Shahidi, Jennifer Padwal, Euyhyun Lee, Ronghui Xu, Sarah Northway, Lissa Taitano, Tiffany Wu, Kamshad Raiszadeh.

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
