## [Decision Letter · Decision Letter 0]

12 Jul 2022

PONE-D-22-15309Factors impacting adherence to an exercise-based physical therapy program for individuals with low back painPLOS ONE

Dear Dr. Shahidi,

Thank you for submitting your manuscript to PLOS ONE. After careful consideration, we feel that it has merit but does not fully meet PLOS ONE’s publication criteria as it currently stands. Therefore, we invite you to submit a revised version of the manuscript that addresses the points raised during the review process.

We look forward to receiving your revised manuscript.

Kind regards,

Shao-Hsien Liu

Academic Editor

PLOS ONE

Journal Requirements:

2. Please ensure you have stated in the Methods section of your manuscript text the full formal name of the institutional review board or ethics committee that approved this study. Please ensure you also state in your manuscript text that 1) the data were analyzed anonymously, and 2) that the review board waived the need for participant consent (as you have stated in the Ethics Statement section of the online submission form).

Reviewers' comments:

Reviewer's Responses to Questions

**Comments to the Author**

1. Is the manuscript technically sound, and do the data support the conclusions?

Reviewer #1: Yes

Reviewer #2: Yes

Reviewer #3: Yes

2. Has the statistical analysis been performed appropriately and rigorously? 

Reviewer #1: Yes

Reviewer #2: Yes

Reviewer #3: Yes

3. Have the authors made all data underlying the findings in their manuscript fully available?

Reviewer #1: No

Reviewer #2: Yes

Reviewer #3: Yes

4. Is the manuscript presented in an intelligible fashion and written in standard English?

Reviewer #1: Yes

Reviewer #2: Yes

Reviewer #3: Yes

5. Review Comments to the Author

Reviewer #1: Thank you for the opportunity to review this important paper. Overall it is an important topic that is often under-examined. However, I have a few major and some minor comments that should be addressed before considering this manuscript for publication.

Major Comments:

1) It would be beneficial to define adherence in the introduction. In the introduction and the abstract, it is unclear if adherence has to do with adhering to the rehabilitation program, which includes attending appointments or adhering to the program provided by the rehab experts, such as home exercise programs or general physical activity recommendations. Part of this is the novelty of this research, most research focuses on adherence to HEP and not adherence to attending PT sessions, so the readers can get easily confused.

2) In the introduction, you mention logistic limitations, such as lack of insurance coverage. Thus it is highly recommended that insurance coverage is added to the independent predictor variables included in your model. It is also recommended that lack of geographic accessibility is incorporated into the models as they are the top factors you identified as being contributors to non-adherence.

3) Most of the discussion are the results written in a narrative format. It is recommended that the authors expand on the discussion to add insight into the "so what?"

Minor Comments:

Abstract:

1) Your objective is related to predictors of adherence and reasons for non-adherence. But your results are primarily about those who discontinued the program. Recommend combining the first four sentences and expanding on the last one, which relates more to the objectives.

2) The conclusion should relate to your results. I think re-writing your results will help with this.

Introduction:

1) Some more background information about potential factors/expanding on the following: "Logistic limitations such as lack of insurance coverage and accessibility restrictions have been shown to affect trial enrollment and may reduce selection of populations with restricted resources." This would also help justify the reason you picked the four discharge categories on top of page 8.

Methods:

1) line 116-please specify wat clinics, outpatient physical therapy clinics?

2) It would be valuable to justify your independent predictors in the introduction or in the methods. Currently, most of the predictor variables seem random. Furthermore, some of the predictors included in the model per Table 2 are not presented in the methods. Including these in the methods would decrease the appearance of the predictors being at random.

3) Since chart data is used, it would be valuable to have a section on how missingness will be/was handled.

Results:

1) Please add a footnote on abbreviations used in the table.

2) Table 1- I find it hard to believe that Baseline VAS is significant when the 95%CI includes 1. Please double-check these values.

Discussion:

1) I fully believe that this is novel and understudied, but I can't entirely agree with the second statement as a quick google search found the following article and results:

https://bmcmusculoskeletdisord.biomedcentral.com/articles/10.1186/1471-2474-11-124

“actors that significantly correlated with adherence included: age (r = 0.7, p < 0.05), initial pain intensity (r = 0.5, p < 0.05), and subjective report of improvement (r = 0.7, p < 0.01). Adherence did not correlate with the type of LBP, patient occupation, experience or nationality of the physiotherapist."

2) I would like to see references or further support regarding the conclusion written on line 259, "suggesting adherence is not affected by insufficient health capacity."

3) The information in paragraph three of the discussion strengthens the reasons why this study should be conducted but doesn't expand on the results. I recommend moving most of this information into the introduction.

4) Throughout the paper, you mention supervised sessions, but this is different from skilled physical therapy sessions. At some point, this needs to be differentiated. Are these patients receiving skilled physical therapy or participating in a supervised exercise program? I say this because many of the references you are using in the discussion are from supervised exercise programs and not skilled physical therapy, and adherence to these are expected to be different.

Reviewer #2: Reviewer Summary and General Comments

This manuscript assess demographic and specific clinical factors as predictors of adherence to an exercise-based rehabilitation program among individuals with low back pain. In addition, they assess reasons individuals were non-adherent to the program. The contributions of these findings to the literature are very important given the lack of adherence assessment within the literature, as well as lack of assessing why individuals struggle to adhere to these important and effective rehabilitation programs. However, there are a few minor points that are unclear to the reviewer and should be addressed prior to the publication of this manuscript. Please see the list below by each section:

Ethics Statement

• It is not stated within this statement whether consent was written or oral. Please clarify.

Abstract

• Check spacing after periods throughout. It looks like a single space in generally used, however within the conclusion on Line 47 there appear to be extra spaces.

Introduction

• Line 71-72 – Do you have a reference for this statement? In addition, do the authors have information as to how often recurrences and progression to chronic LBP occur?

Methods

• Line 115 – As mentioned above, can you clarify whether written/oral informed consent was obtained? Additionally, was the IRB through UCSD? If so, please include here as well.

• Line 117-118 – The dates as currently written here are a bit confusing to read, may be clearer to write out the month and then year (e.g., November 2015).

• Line 118 – Can the authors provide a rationale for using gender instead of the more appropriate term sex? Or possibly clarify individuals who identified as either male or female gender had information collected?

• Lines 126-137 – This section may benefit from numbering the criteria for each specific diagnostic category. For example: “… patients were categorized in the lumbar stenosis category if they had, 1) neurogenic claudication and/or radicular leg symptoms, or 2) confirmatory cross-sectional imaging showing lumbar spinal stenosis at one or more levels.”

• Lines 139-140 – Can the author clarify how long the rehabilitation program was within this sentence? It appears it was 10 weeks, but the exact duration is unclear.

• Although mentioned later in the manuscript, a brief explanation for how missing data were handled in the statistical analysis section should be included.

• Line 175 – Can the authors define NSAID. This appears to be the first time NSAID has been written in the paper and thus should be defined.

• Can the authors provide a rationale for not looking at adherence as a continuous measure (e.g., percent adherence), in addition to the binary measure used in this analysis?

Results/Tables

• Check alignment/spacing within each table and table legend.

• A footnote should be included for each table that should include the definition of each abbreviation used within the table.

• Table 1 – the colons used for gender, smoking history, diabetes diagnosis, and radiculopathy diagnosis are a bit confusing. Perhaps a / would be clearer to use. (e.g., 44.8/55.2)

• Table 2 – not all units are included within this table, please check to be sure they are added for each variable.

Discussion

• Line 248 – Given this is a rehabilitation program, not necessarily a supervised exercise program, this line should be adjusted to read along the lines of: “..long term supervised rehabilitation exercise program…”

• Lines 260-263 – Did the authors happen to examine the demographic breakdown of those who reported discontinuation due to inability to obtain insurance authorization or transportation means? May be something important to include depending on the results.

• Lines 291-297 – Perhaps including the authors thoughts as to why such a big difference in percent adherence was found between the two example programs mentioned may add to the impact of this paragraph.

Conclusion

• Line 336-337 – The authors should re-word, there appears to be a missing word or some re-arranging that makes this sentence unclear.

Reviewer #3: 1. In their investigation of adherence, the authors do not consider activity tolerance to the prescribed intervention as a factor in adherance. Consideration for pain rating and pain trajectory may strengthen the argument for tolerance and appropriateness of the exercise prescription provided to the study group.

2. The authors do not detail why BMI was an exclusion criteria. Elaboration of this point would help justification of the sample.

6. PLOS authors have the option to publish the peer review history of their article (what does this mean?). If published, this will include your full peer review and any attached files.

Reviewer #1: No

Reviewer #2: **Yes: **Katherine Ann Collins

Reviewer #3: **Yes: **Jenna M. Tosto-Mancuso

---

## [Author Response · Author response to Decision Letter 0]

16 Aug 2022

We appreciate the reviewers comments and suggestions. We have incorporated responses and revisions according to the reviewer requests and feel that the manuscript is greatly improved as a result. We have included our responses to the reviewer comments below in-line. 

Reviewer #1: Thank you for the opportunity to review this important paper. Overall it is an important topic that is often under-examined. However, I have a few major and some minor comments that should be addressed before considering this manuscript for publication.

Major Comments:

1) It would be beneficial to define adherence in the introduction. In the introduction and the abstract, it is unclear if adherence has to do with adhering to the rehabilitation program, which includes attending appointments or adhering to the program provided by the rehab experts, such as home exercise programs or general physical activity recommendations. Part of this is the novelty of this research, most research focuses on adherence to HEP and not adherence to attending PT sessions, so the readers can get easily confused.

Thank you for this comment. Given that this is a study of a supervised in-clinic physical therapy program, we are considering adherence primarily based on attendance to those in-clinic visits. Beyond adherence through prescribed attendance, we recognize that compliance to any non-supervised components of the treatments, such as home exercises, was not addressed or monitored in this study. This definition has been clarified in the methods and the limitations related to not collecting compliance information is added as a limitation in the discussion section. 

2) In the introduction, you mention logistic limitations, such as lack of insurance coverage. Thus it is highly recommended that insurance coverage is added to the independent predictor variables included in your model. It is also recommended that lack of geographic accessibility is incorporated into the models as they are the top factors you identified as being contributors to non-adherence.

Thank you for this comment. Unfortunately, we were not able to retrieve accurate insurance coverage data for this cohort of patients. One limitation to our data is that the data on reasons for discontinuation of treatment was acquired via anonymous survey response and is not able to be aligned to the origin payor data on a per-subject basis for which adherence was modeled. Because of this, we are unable to adjust/include the insurance or geographic accessibility limitations as an independent variable in the model. We have included this as a limitation in the discussion. 

3) Most of the discussion are the results written in a narrative format. It is recommended that the authors expand on the discussion to add insight into the "so what?"

Thank you for this comment. We have now expanded on the discussion to include an emphasis on the clinical impact of these data, and proposed action items based on these findings. 

Minor Comments:

Abstract:

1) Your objective is related to predictors of adherence and reasons for non-adherence. But your results are primarily about those who discontinued the program. Recommend combining the first four sentences and expanding on the last one, which relates more to the objectives.

Thank you. We have now redistributed the focus of the abstract to be more aligned with the results. 

2) The conclusion should relate to your results. I think re-writing your results will help with this.

Thank you. We have modified the conclusion to align more closely with the results. 

Introduction:

1) Some more background information about potential factors/expanding on the following: "Logistic limitations such as lack of insurance coverage and accessibility restrictions have been shown to affect trial enrollment and may reduce selection of populations with restricted resources." This would also help justify the reason you picked the four discharge categories on top of page 8.

Thank you. We have added additional background emphasizing the identification of logistic limitations as being impactful in trials and treatments, however our discharge categories were selected based on the most commonly observed reasons for non-adherence, as opposed to being selected on an a-priori basis. As such, we aimed to encompass the most similar barriers together. We did not observe large variability in reasons for non-adherence or discharge within these categories. We have clarified this in the methods section. 

Methods:

1) line 116-please specify what clinics, outpatient physical therapy clinics?

Thank you. We have now specified that these treatments took place in outpatient physical therapy clinics. 

2) It would be valuable to justify your independent predictors in the introduction or in the methods. Currently, most of the predictor variables seem random. Furthermore, some of the predictors included in the model per Table 2 are not presented in the methods. Including these in the methods would decrease the appearance of the predictors being at random.

These variables were based on both evidence from prior studies indicating that these variables are potentially impactful in clinical outcomes (e.g. age, BMI, smoking history, diabetes, diagnosis, pain, disability and quality of life scores), as well as more granular detail on muscular performance based on the exercise-based focus of the clinical rehabilitation program (Baseline exercise strength level). We have now included this selection rationale in the methods section along with supporting references for the impact of these variables on clinical outcomes. We acknowledge that other factors, such as psychosocial factors, have been shown to influence clinical outcomes and were not available for collection within this data set. We acknowledge this as a limitation in our study. 

3) Since chart data is used, it would be valuable to have a section on how missingness will be/was handled.

Results:

Because there were some variables with missing data, we did not remove a row completely from the analysis if the row contained any missing values. Instead, we removed rows only when a variable that was used for the specific tests were missing (pairwise deletion). We did not perform imputation to fill the missing values (no replacement). As such, the multivariate regression model includes only patients with complete data (N size (2,181) now included in regression table). This is now clarified in the methods section. 

1) Please add a footnote on abbreviations used in the table.

A footnote is now included for tables with abbreviations

2) Table 1- I find it hard to believe that Baseline VAS is significant when the 95%CI includes 1. Please double-check these values.

This is an artifact of rounding for the number of significant digits reported in the table; The confidence interval was extremely small for VAS (0.992-0.999). We have modified the significant digits to clarify this. 

Discussion:

1) I fully believe that this is novel and understudied, but I can't entirely agree with the second statement as a quick google search found the following article and results:

https://bmcmusculoskeletdisord.biomedcentral.com/articles/10.1186/1471-2474-11-124

“actors that significantly correlated with adherence included: age (r = 0.7, p < 0.05), initial pain intensity (r = 0.5, p < 0.05), and subjective report of improvement (r = 0.7, p < 0.01). Adherence did not correlate with the type of LBP, patient occupation, experience or nationality of the physiotherapist."

Thank you for bringing this citation to our attention. We agree that this article does provide specific predictors to musculoskeletal physical therapy in a quantitative manner, however the methods do not specify whether the physical therapy was primarily focused on exercise-based modalities as is the focus of the current study. Regardless, we agree that this should be recognized and now have included this citation in the discussion. 

2) I would like to see references or further support regarding the conclusion written on line 259, "suggesting adherence is not affected by insufficient health capacity."

This comment was made in reference to the observed results and was intended to communicate that a very low proportion of participants reported that they were unable to complete the program due to unrelated medical issues. Our interpretation of this is that severe medical comorbidities were not a limiting factor to program adherence in this cohort. We have clarified this in the discussion section. 

3) The information in paragraph three of the discussion strengthens the reasons why this study should be conducted but doesn't expand on the results. I recommend moving most of this information into the introduction.

Thank you. We have moved this section to the introduction and abridged for conciseness.

4) Throughout the paper, you mention supervised sessions, but this is different from skilled physical therapy sessions. At some point, this needs to be differentiated. Are these patients receiving skilled physical therapy or participating in a supervised exercise program? I say this because many of the references you are using in the discussion are from supervised exercise programs and not skilled physical therapy, and adherence to these are expected to be different.

The rehabilitation program evaluated in this study was an exercise-based physical therapy program administered by a licensed physical therapist. This has been clarified in the methods section. 

Reviewer #2: Reviewer Summary and General Comments

This manuscript assess demographic and specific clinical factors as predictors of adherence to an exercise-based rehabilitation program among individuals with low back pain. In addition, they assess reasons individuals were non-adherent to the program. The contributions of these findings to the literature are very important given the lack of adherence assessment within the literature, as well as lack of assessing why individuals struggle to adhere to these important and effective rehabilitation programs. However, there are a few minor points that are unclear to the reviewer and should be addressed prior to the publication of this manuscript. Please see the list below by each section:

Ethics Statement

• It is not stated within this statement whether consent was written or oral. Please clarify.

We have now clarified that a waiver of consent was obtained due to the retrospective and de-identified nature of the data. 

Abstract

• Check spacing after periods throughout. It looks like a single space in generally used, however within the conclusion on Line 47 there appear to be extra spaces.

Thank you. We have double checked and corrected inconsistent spacing throughout. 

Introduction

• Line 71-72 – Do you have a reference for this statement? In addition, do the authors have information as to how often recurrences and progression to chronic LBP occur?

We have now included additional references to support this statement as well as additional information on prevalence of recurrence and LBP progression.

Methods

• Line 115 – As mentioned above, can you clarify whether written/oral informed consent was obtained? Additionally, was the IRB through UCSD? If so, please include here as well.

Thank you. We have now clarified the location and modality of consent in the methods section. 

• Line 117-118 – The dates as currently written here are a bit confusing to read, may be clearer to write out the month and then year (e.g., November 2015).

Thank you, we have modified the dates to improve clarity.

• Line 118 – Can the authors provide a rationale for using gender instead of the more appropriate term sex? Or possibly clarify individuals who identified as either male or female gender had information collected?

We did not collect non-binary gender or gender identity information, and as such have changed our terminology to refer to sex instead of gender. 

• Lines 126-137 – This section may benefit from numbering the criteria for each specific diagnostic category. For example: “… patients were categorized in the lumbar stenosis category if they had, 1) neurogenic claudication and/or radicular leg symptoms, or 2) confirmatory cross-sectional imaging showing lumbar spinal stenosis at one or more levels.”

Thank you, we have now organized our diagnostic categories using numbering as suggested. 

• Lines 139-140 – Can the author clarify how long the rehabilitation program was within this sentence? It appears it was 10 weeks, but the exact duration is unclear.

The prescribed program was 20 visits, with a frequency of 2 visits per week (10 weeks). This is indicated in the Exercise Protocol subsection of the methods.

• Although mentioned later in the manuscript, a brief explanation for how missing data were handled in the statistical analysis section should be included.

We have now included that we used pairwise deletion to handle missing data in the statistical analysis section. 

• Line 175 – Can the authors define NSAID. This appears to be the first time NSAID has been written in the paper and thus should be defined.

Thank you. NSAIDs refer to Non-Steroidal Anti-inflammatory Drugs. We have now defined this acronym in the text. 

• Can the authors provide a rationale for not looking at adherence as a continuous measure (e.g., percent adherence), in addition to the binary measure used in this analysis?

Because the treatment was a prescribed program that was based on the minimum estimated training prescription required to elicit physiological adaptation of muscle (6-7 weeks at 2 times/week), we based our adherence measure according to this recommended prescription. Analytically, it is feasible to treat adherence as a continuous measure, however the impact of adherence on programs of a longer duration that are sufficient to elicit physiological adaptation of muscle was a primary focus (and novelty) of this experiment. However, we acknowledge that this approach may be useful in understanding key timepoints at which patients may be dropping out of the program and may be of interest in future studies. 

Results/Tables

• Check alignment/spacing within each table and table legend.

Thank you, we have now double checked and corrected for spacing inconsistencies within the tables/legends.

• A footnote should be included for each table that should include the definition of each abbreviation used within the table.

We have now included footnotes with abbreviation definitions in the tables. 

• Table 1 – the colons used for gender, smoking history, diabetes diagnosis, and radiculopathy diagnosis are a bit confusing. Perhaps a / would be clearer to use. (e.g., 44.8/55.2)

We have modified the table to clarify these data as suggested. 

• Table 2 – not all units are included within this table, please check to be sure they are added for each variable.

We have now included units for all variables in Table 2. 

Discussion

• Line 248 – Given this is a rehabilitation program, not necessarily a supervised exercise program, this line should be adjusted to read along the lines of: “..long term supervised rehabilitation exercise program…”

Thank you. We have modified this statement to clarify that this is a supervised exercise-based rehabilitation program. 

• Lines 260-263 – Did the authors happen to examine the demographic breakdown of those who reported discontinuation due to inability to obtain insurance authorization or transportation means? May be something important to include depending on the results.

We agree that the demographic breakdown of those who reported different logistical considerations associated with their discontinuation is interesting, however, because in our data reasons for discontinuation were anonymously collected, we are not able to connect the patient characteristics to the reasons for non-adherence. As such, we are unable to determine if there is an association between demographic features and insurance authorization/transportation. 

• Lines 291-297 – Perhaps including the authors thoughts as to why such a big difference in percent adherence was found between the two example programs mentioned may add to the impact of this paragraph.

Thank you for this suggestion. As this paragraph has been substantially modified and moved to the introduction as recommended by reviewer 1, the context of the information here has been shifted to focus on gaps in the literature as opposed to interpretation of differences in adherence rates across modalities of therapy. We did not expand on this concept in order to improve flow and logic of the introduction in its modified form.

Conclusion

• Line 336-337 – The authors should re-word, there appears to be a missing word or some re-arranging that makes this sentence unclear.

Thank you. We have attempted to clarify the conclusion based on this and the previous reviewers’ comments. 

Reviewer #3: 1. In their investigation of adherence, the authors do not consider activity tolerance to the prescribed intervention as a factor in adherence. Consideration for pain rating and pain trajectory may strengthen the argument for tolerance and appropriateness of the exercise prescription provided to the study group.

We agree that a potential mismatch between the prescribed program and the activity tolerance of the individual may impact adherence. Although we do not have specific data on activity tolerance in this sample, the program was supervised by a licensed physical therapist that is trained to identify appropriate prescription of exercise intensity based on the tolerance and physiological capacity of the individual in the context of their individual pathology and symptomology. However, we agree that patients with high pain levels and trajectories may be differentially adherent, and we have a manuscript in preparation evaluating the clinical outcomes (including pain trajectories) of patients participating in this program. Although we did not observe significant differences in back strength at baseline between groups, we did observe that the baseline pain scores for individuals who did not complete the program were significantly higher than those who did (2-point difference). However, this difference is 10 times less than that considered to be clinically significant (20-point difference). As such, it is unlikely that differences in pain were a significant driver for lack of adherence. Interestingly we observed, through a coarse evaluation of pain change per visit over the duration of attended treatments, that those who were not adherent actually had larger improvements in symptoms on a per-visit basis compared to those who were adherent. This may suggest that some patients who were not adherent experienced rapid recovery and ceased attending due to improvement in symptoms. This is also supported by our data demonstrating that a proportion of non-adherent patients experienced improvements, but never underwent formal discharge. Because we are concurrently preparing a manuscript evaluating clinical outcome trajectories and associated predictors, we have not included a detailed discussion of these observations in the current submission. 

2. The authors do not detail why BMI was an exclusion criteria. Elaboration of this point would help justification of the sample.

Only patients with BMI values that were observed to be non-physiological for the age-range studied were excluded from the analysis. Based on the numeric values observed for those excluded, we hypothesize that these patients had incorrectly entered BMI data and therefore those values were not retained in analysis. We have clarified this in the patient characteristics section.

---

## [Decision Letter · Decision Letter 1]

5 Oct 2022

Factors impacting adherence to an exercise-based physical therapy program for individuals with low back pain

PONE-D-22-15309R1

Dear Dr. Bahar Shahidi,

We’re pleased to inform you that your manuscript has been judged scientifically suitable for publication and will be formally accepted for publication once it meets all outstanding technical requirements.

Kind regards,

Ravi Shankar Yerragonda Reddy, Ph.D

Academic Editor

PLOS ONE

Reviewers' comments:

Reviewer's Responses to Questions

**Comments to the Author**

1. If the authors have adequately addressed your comments raised in a previous round of review and you feel that this manuscript is now acceptable for publication, you may indicate that here to bypass the “Comments to the Author” section, enter your conflict of interest statement in the “Confidential to Editor” section, and submit your "Accept" recommendation.

Reviewer #2: All comments have been addressed

2. Is the manuscript technically sound, and do the data support the conclusions?

Reviewer #2: Yes

3. Has the statistical analysis been performed appropriately and rigorously? 

Reviewer #2: Yes

4. Have the authors made all data underlying the findings in their manuscript fully available?

Reviewer #2: Yes

5. Is the manuscript presented in an intelligible fashion and written in standard English?

Reviewer #2: Yes

6. Review Comments to the Author

Reviewer #2: The authors have done a nice job with this revision and have addressed all of my comments. Therefore, I have no further comments.

7. PLOS authors have the option to publish the peer review history of their article (what does this mean?). If published, this will include your full peer review and any attached files.

Reviewer #2: **Yes: **Katherine A. Collins

---

## [Editor Report · Acceptance letter]

13 Oct 2022

PONE-D-22-15309R1 

Factors impacting adherence to an exercise-based physical therapy program for individuals with low back pain 

Dear Dr. Shahidi:

I'm pleased to inform you that your manuscript has been deemed suitable for publication in PLOS ONE. Congratulations! Your manuscript is now with our production department. 

Kind regards, 

on behalf of

Dr. Ravi Shankar Yerragonda Reddy 

Academic Editor

PLOS ONE